Role of contrast-enhanced ultrasound with time-intensity curve analysis about thyroid nodule and parenchyma for differentiating BRAF V600E mutation status

Hu Zhipeng 1
Xue Rong 1
Liu Zhixi 2
Liu Liang 1
Gong Zheli 1 109398375@qq.com
1 Department of Ultrasound Medicine, Hunan Provincial People’s Hospital, The First Affiliated Hospital of Hunan Normal University , Changsha, Hunan , China
2 Department of Social Medicine, Hunan Provincial People’s Hospital, The First Affiliated Hospital of Hunan Normal University , Changsha, Hunan , China
Yapıcı Sercan
Electronic publication date: 2025 Feb 25
Publication date: 2025
Volume: 13
Electronic Location ID: e19006
Received 2024 Nov 19; Accepted 2025 Jan 27
Copyright: © 2025 Hu et al.
Copyright year: 2025
Copyright holder: Hu et al.
License: This is an open access article distributed under the terms of the Creative Commons Attribution License, which permits unrestricted use, distribution, reproduction and adaptation in any medium and for any purpose provided that it is properly attributed. For attribution, the original author(s), title, publication source (PeerJ) and either DOI or URL of the article must be cited.
License URL: https://creativecommons.org/licenses/by/4.0/

Keywords: Contrast enhanced ultrasound, Quantitative parameter, BRAF V600E, Papillary thyroid microcarcinoma

Funding: Natural Science Foundation of Hunan Province 2023JJ40373 Scientific Research Projects of Hunan Provincial Health Commission B202309026144 This work was supported by the Natural Science Foundation of Hunan Province (Grant No. 2023JJ40373), Scientific Research Projects of Hunan Provincial Health Commission (Grant No. B202309026144). The funders had no role in study design, data collection and analysis, decision to publish, or preparation of the manuscript.

==============================
Background

The BRAF V600E mutation was proven associated with papillary thyroid cancer (PTC) which has more aggressive behavior and could affect the outcome of PTC. The objective of this study was to observe more contrast-enhanced ultrasound (CEUS) time-intensity curve (TIC)—based quantitative parameters in nodules and surrounding parenchyma and analyze the association between the TIC-based quantitative parameters and BRAF V600E mutation status in patients with PTC.

Methods

A retrospective analysis of 447 PTC patients was conducted. Prior to thyroidectomy or fine needle aspiration (FNA), all patients had CEUS and had their BRAF V600E mutations examined. Based on their mutation status, the patients were split into two groups. The two groups were compared in terms of sex, age, quantitative CEUS characteristics, pathological findings, vascular invasion, capsular invasion, and cervical lymph node metastases.

Results

A total of 240 patients were in the mutation negative group and 207 patients were in the BRAF mutation positive group. The BRAF-positive group exhibited significantly higher arrival time (AT) and time to peak enhancement (TTP) of the nodules, among other direct quantitative characteristics. The BRAF-positive mutant nodules showed significantly higher arrival time change and time to peak change compared to the surrounding parenchyma for indirect quantitative metrics.

Conclusion

The time-dependent quantitative parameters of CEUS time intensity curve in nodules and surrounding parenchyma have clinical value in distinguishing BRAF V600E mutation positive nodules from gene mutation negative nodules. Quantitative CEUS characteristics may be beneficial in detecting the BRAF V600E mutation status and informing the subsequent clinical choice.

Introduction

Papillary thyroid carcinoma (PTC) is the most common endocrine tumor and accounts for approximately 80–85% of malignant tumors of thyroid (Gharib et al., 2013; Ali, 2011). The conventional PTC treatment was surgery, radioactive iodine, thyroid hormone suppression of TSH, external beam radiation (less commonly), and rarely, chemotherapy (Luster et al., 2019). While the majority of PTCs have a low death rate and respond well to radioiodine therapy, certain PTCs have worse prognosis and more aggressive clinicopathologic features (Feng et al., 2019; Yamada et al., 2014). Recently, biological differences between PTCs have been discovered from the identification of genetic alterations in various signaling pathways (Lee et al., 2017). One of the genetic mutations linked to the protein kinase pathway is the B-type Raf (BRAF) V600E kinase mutation on exon 15 (Riesco-Eizaguirre & Santisteban, 2016; Poulikakos, Sullivan & Yaeger, 2022). This mutation has been linked to thyroid cancer by upregulating cell division and proliferation, which results in the development of tumors (Chen et al., 2021; Fu et al., 2023). Additionally, numerous studies have demonstrated that the BRAF V600E mutation is linked to the invasive clinicopathological behavior of PTC as well as negative outcomes such distant metastases, lymph node metastases, loco-regional recurrences, and malignant recurrences (Haoran & Yongzhao, 2017; Yong-Heng et al., 2018; Jin et al., 2017; Baloch et al., 2022). Although the BRAF V600E mutant status is helpful for clinical decision-making, most papillary thyroid microcarcinoma are not recommended for its acquisition since it requires invasive diagnostic procedures such surgery and fine-needle aspiration biopsies (Haugen et al., 2015).

Contrast-enhanced ultrasound (CEUS) is a supplemental ultrasound imaging technique that is frequently employed in clinical settings to detect benign and malignant thyroid nodules. It is also capable of rapidly predicting the severity and prognosis of PTC (Kim et al., 2019; Ricarte-Filho et al., 2012). The advantage of CEUS is that it can measure more precisely and accurately in terms of shape. It can also evaluate the thyroid nodules’ hemodynamics and the sequence and intensity of vascular perfusion based on several parameters (washout, washout peak, ring sign, etc.) (Trimboli et al., 2020). According to certain research, CEUS has a strong predictive value for detecting PTC (Diao et al., 2017; Xi et al., 2020; Yan et al., 2018), but most studies were focused on the contrast enhancement patterns (Zhang et al., 2020; Fang et al., 2021; Wang et al., 2021). The BRAF V600E mutation was linked to certain CEUS quantitative characteristics in a number of investigations, although these research only examined the nodular parameters obtained by post-processing software, like arrival time (AT) and time to peak (TTP) (Chen et al., 2019). In this study, we aimed to compare both the nodular parameters and the quantitative parameters of the thyroid nodule and surrounding parenchyma in relation to BRAF status. This study is expected to provide additional evidence for differentiating aggressive PTC and to assist in personalized clinical treatment.

Materials and Methods

Study subjects

This study was approved by Medical Ethics Committee of Hunan Provincial People’s Hospital (The First Affiliated Hospital of Hunan Normal University) (IRB Approval No: [2024]-261). PTC nodules receiving CEUS scans at our hospital were found by looking through the CEUS imaging database. A total of 447 PTC nodules that were pathologically confirmed by surgery or FNA between October 2021 to December 2023 were included in the study. A total of 240 individuals were classified as BRAF-negative, while 207 patients with a BRAF V600E mutation were classified as BRAF-positive. Exclusion criteria included: (1) absence of preoperative CEUS; (2) absence of pathological diagnosis; (3) unquantified CEUS pictures; and (4) refusal to use clinical data.

CEUS study protocol

Two radiologists with more than 5 years of CEUS experience performed the CEUS scans. The other two radiologists with more than 7 years of CEUS experience reviewed the CEUS data. When a patient had more than one nodule, the target nodule was chosen based on its level of suspicion. The Mindray Resona 7 (Mindray Bio-Medical Electronics Co., Ltd., Shenzhen, China) or LOGIQ E9 ultrasound scanner (GE Healthcare, Wauwatosa, WI, USA) with convex array probes of 5–15 MHz were used for all ultrasonographic investigations. A low mechanical index (MI < 0.10) was maintained at all times to reduce artificial signal and microbubble destruction. The focal zone was always placed at the nodule’s base for comparison. A 5 mL saline flush (0.9% sodium chloride) was administered after a 1.5 mL intravenous bolus of contrast agent (SonoVue, Bracco International, Milan, Italy). Every nodule underwent a minimum of 120 s of dynamic scanning. For future examination, every CEUS image was saved on the hardware.

Image analysis

Two radiologists with over 5 years of CEUS expertise examined the quantitative CEUS parameters of PTC nodules. Both the nodules and the surrounding thyroid parenchyma were designated as regions of interest (ROI). The ultrasonic signal augmentation status was quantitatively examined using time-intensity curve (TIC) analysis. The following direct quantitative parameters were measured (Fig. 1): (1) The arrival time, AT; (2) time to peak, TTP; (3) basic intensity, BI; (4) peak intensity, PI. In addition, we measured more parameters which based on the direct quantitative parameters: (1) rise time, RT; (2) arrival time change between nodule and surrounding parenchyma, ATC; (3) time to peak change between nodule and surrounding parenchyma, TTPC; (4) rise time ratio, RTR; (5) time to peak ratio, TTPR; (6) basic intensity change between nodule and surrounding parenchyma, BIC; (7) peak intensity change between nodule and surrounding parenchyma, PIC; (8) rise intensity, RI; (9) peak intensity ratio, PIR; (10) rise intensity ratio, RIR.

Figure 1 Schematic of time-intensity curve, including all parameters.

Pathological examination and BRAF V600E mutation analysis

All surgical specimens were classified by skilled pathologists who were blind to the patient’s medical history and ultrasonographic results in accordance with the World Health Organization’s 2003 histological categorization of thyroid cancers (Delellis, Lloyd & Heitz, 2004). Pathological diagnosis was considered as the gold standard for the study.

The BRAF V600E mutation analysis was performed at the Pathology Department of the Hunan Provincial People’s Hospital, The First Affiliated Hospital of Hunan Normal University in China. Following the manufacturer’s instructions, the DNA of surgically obtained tissue samples was extracted using a Qiagen QIA amp DNA FFPE tissue kit (56404; Qiagen, Hilden, Germany) in 50mL of buffer ATE (a component of the kit). Using a Merinton SMA 4000 spectrophotometer (Merinton Inc., Beijing, China), the absorbance of the isolated DNA was measured. The polymerase chain reaction was used to amplify BRAF exon 15 (forward: TCATAATGCTTGCTCTGATAGGA, reverse: GGCCAAAAATTTAA TCAGTGGA).

Statistical analysis

The SPSS 29.0 statistical software (SPSS Inc., Chicago, IL, USA) was used. Quantitative data with an abnormal distribution were evaluated using the Mann-Whitney U test and were referred to as the median (interquartile range). The Pearson chi-square test was used to assess the categorical data, which were denoted by N (%). p value < 0.05 was considered statistically significant.

Results

This study included 447 individuals (203 men and 244 women) who were enrolled between October 2021 and December 2023. Table 1 displays the clinical features of the tumors and patients. Multiple lesions, capsular invasion, and lymph node metastases did not significantly correlate with the presence of a BRAF mutation in our study.

Table 1 Demographic and clinical characteristics of the patients.

Features	BRAF V600E	p valuea	
(+)	(−)	
Age, y	47.2 (33.8, 55.8)	48.7 (34.8, 57.2)	0.474b	
Sex			0.057	
Male	84 (40.6)	119 (49.6)		
Female	123 (59.4)	121 (50.4)		
Multiple lesions			0.129	
Absent	155 (74.9)	194 (80.8)		
Present	52 (25.1)	46 (19.2)		
Capsular invasion			0.207	
Absent	127 (61.4)	161 (67.1)		
Present	80 (38.6)	79 (32.9)		
Vessel invasion			0.666	
Absent	204 (98.6)	238 (99.2)		
Present	3 (1.4)	2 (0.8)		
Lymph node metastasis			0.126	
Absent	139 (67.1)	117 (73.8)		
Present	68 (32.9)	63 (26.3)		
Notes:

Values are presented as median (interquartile range) or numbers (%).

a Chi-square test.

b Mann-Whitney U test.

The quantitative CEUS parameters were given using TIC curves. Direct measures showed that nodules in the BRAF-positive group had longer AT (7.68 (5.96, 8.99)) and TTP (15.58 (13.97, 17.42)) than nodules in the BRAF-negative group (6.82 (5.73, 8.48)) and TTP (14.62 (12.56, 17.52)) (p = 0.019 and p = 0.023, respectively). There was no correlation between the BI and PI and the state of mutations (Table 2).

Table 2 Comparison of direct quantitative TIC parameters between BRAF V600E mutation positive/negative PTC nodules.

Parameters	BRAF V600E	p value	
(+)	(−)	
AT, s	7.68 (5.96, 8.99)	6.82 (5.73, 8.48)	0.019*	
TTP, s	15.58 (13.97, 17.42)	14.62 (12.56, 17.52)	0.023*	
BI, dB	19.67 (16.94, 21.03)	18.53 (17.44, 20.73)	0.336	
PI, dB	27.71 (25.84, 29.76)	27.35 (25.33, 29.68)	0.445	
Notes:

The data was presented as median (interquartile range). PTC, papillary thyroid carcinoma; AT, arrival time; TTP, time to peak; BI, basic intensity; PI, peak intensity.

* p value < 0.05.

In terms of indirect parameters, ATC (3.80 (2.33, 5.30)) and TTPC (4.07 (3.17, 4.61)) in the BRAF-positive group were longer than ATC (3.13 (2.36, 4.14)) and TTPC (3.85 (3.16, 4.31)) in the BRAF-negative group (p = 0.002 and p = 0.036 respectively), while other indirect parameters differences between the two groups were not statistically significant (Table 3).

Table 3 Comparison of indirect quantitative TIC-based parameters of BRAF V600E mutation positive/negative PTC nodules and surrounding thyroid parenchyma.

Parameters	BRAF V600E	p value	
(+)	(−)	
RT, s	8.00 (6.57, 9.48)	7.57 (5.61, 10.02)	0.121	
ATC, s	3.80 (2.33, 5.30)	3.13 (2.36, 4.14)	0.002*	
TTPC, s	4.07 (3.17, 4.61)	3.85 (3.16, 4.31)	0.036*	
RTR	1.00 (0.84, 1.22)	1.08 (0.91, 1.26)	0.078	
TTPR	1.33 (1.25, 1.43)	1.35 (1.25, 1.44)	0.451	
BIC, dB	−2.26 (−3.37, −1.68)	−2.35 (−3.19, −1.56)	0.864	
PIC, dB	−3.21 (−4.69, −2.41)	−3.42 (−4.90, −2.52)	0.485	
RI, dB	8.53 (6.98, 9.87)	8.52 (7.08, 9.85)	0.897	
PIR	0.89 (0.85, 0.92)	0.89 (0.85, 0.92)	0.448	
RIR	0.91 (0.76, 1.05)	0.88 (0.76, 1.02)	0.443	
Notes:

The data was presented as median (interquartile range). RT, rise time; ATC, arrival time change between nodule and surrounding parenchyma; TTPC, time to peak change between nodule and surrounding parenchyma; RTR, rise time ratio; TTPR, time to peak ratio; BIC, basic intensity change between nodule and surrounding parenchyma; PIC, peak intensity change between nodule and surrounding parenchyma; RI, rise intensity; PIR, peak intensity ratio; RIR, rise intensity ratio.

* p value < 0.05.

Discussion

Over the past few decades, the prevalence of thyroid cancer, one of the most prevalent endocrine cancers, has steadily climbed globally (Miranda-Filho et al., 2021; Kitahara & Sosa, 2020). Although papillary thyroid carcinoma (PTC), the most prevalent histological form of differentiated thyroid cancer, is typically a slow-growing disease, the majority of PTC patients nonetheless underwent surgery in the clinic (Lee & Shin, 2014; Li, Dal Maso & Vaccarella, 2020). The potential complications from surgery including recurrent laryngeal nerve injury, hypoparathyroidism and lifelong thyroid hormone replacement. Considering the comparatively stable incidence-based mortality rates but the higher incidence rates of detection, customized treatments have been proposed (Xing, Haugen & Schlumberger, 2013; Miccoli, 2014; White et al., 2020). Previous studies have verified that patients with PTC frequently carry the BRAF mutation, with prevalence rates reaching 73.4% (Hiroyuki et al., 2003; Maaitah et al., 2019; Xing et al., 2015), while aggressive behaviors, such as capsule invasion, vessel invasion and lymph node metastasis are linked to the BRAF mutation (Conzo et al., 2012; Gambardella et al., 2019; Marotta et al., 2016). Thus, knowing the PTC mutate status offers crucial prognostic information for more individual treatment.

By replacing valine (V) at amino acid 600 with glutamic acid (E), the BRAF V600E mutation is created. This constitutively activates the MAPK pathway, which promotes carcinogenesis. Since its discovery, the BRAF V600E has emerged as a promising diagnostic and prognostic biomarker for PTC (Oler & Cerutti, 2010; Silver et al., 2021). High-risk clinicopathological features, tumor recurrence, metastasis, and decreased susceptibility to radioiodine therapy have all been linked to this mutation (Lee, Lee & Kim, 2007; Kim et al., 2012; Tufano et al., 2012). According to reports, the BRAF V600E mutation raises the aggressiveness of the tumor as well as the mortality and recurrence rates of PTC (Xing et al., 2013). According to studies, the BRAF V600E mutation in PTC has been linked to extra-thyroidal extension, multifocality, lymph node metastasis, and advanced TNM staging. Thus, it is crucial to include the BRAF V600E mutant state in the risk stratification management of PTC in order to lower the mortality and recurrence rates.

Although surgery is still the primary treatment for PTC patients, a number of recent studies have suggested that PTC may be eligible for active surveillance and avoidable surgery. Nevertheless, prior to histological evaluation, we were unable to determine whether these patients had any metastases. Noninvasive mutation status determination is crucial for stratified disease management since the BRAF V600E mutant state may be indicative of these possible malignant variables.

Ultrasound has been proved to be vital in evaluating thyroid diseases (Sui et al., 2016). With the advent and development of ultrasound contrast agents, it is now widely accepted that CEUS can provide more sensitive information of the microcirculation inside lesions and useful in differentiating benign and malignant nodules. The enhancement patterns of PTC on CEUS include centripetal mode of entrance, later arrival, heterogeneous hypo-enhancement, and earlier washout. These characters are associated with relevant pathologic mechanisms. Studies showed that tumor cells secrete large amounts of endothelial growth factor, resulting in intra-nodular vascularization. These new blood vessels of the marginal zone are relatively dense, while the center is sparse which might lead to the entrance mode. In addition, because of vascular stenosis or occlusion due to internal microthrombus, blood vessels with low efficacy and incompletely open state (Yuan et al., 2015), complicated collagen degeneration (Zhang et al., 2010), and the existence of arteriovenous fistula, this might lead to a later arrival but earlier washout enhancement mode in the CEUS process. BRAF V600E mutation is an important factor for the aggressive nature of PTC. The possible reason is that BRAF V600E might increase kinase activity and over-activate the mitogen-activated protein kinase pathway. Studies have shown that CEUS may offer qualitative traits associated with BRAF mutations (Ruan et al., 2022; Petrasova et al., 2022; Fan et al., 2024). However, the expertise of the interpreter determines these qualities accurately, therefore more objective quantitative measures are required. In this research, we have incorporated 10 indirect parameters from comparative studies in addition to quantitative data of the surrounding parenchyma that are readily derived from the TIC curve. Significant variations were seen in the time change parameters in this study, such as AT, TTP, ATC, and TTPC. The possible reason is that the BRAF V600E mutation promotes the formation of psammoma body in micro-calcification and associated with interstitial fibrosis which leads to the nodules showing a harder texture and more calcifications. This might result in a slow progression and slowly reaching a peak for the BRAF mutation-positive group, and could explain the results of our study.

Our study did have several drawbacks. First, this is a retrospective analysis. There may be selective bias in tumor selection. The tumor we choose might more readily available for collection and genetic analysis. A prospective well-designed study is needed for a more accurate assessment of the relation between the CEUS features and BRAF mutations. Second, choosing the point on the TIC to acquire quantitative parameters may result in random error. Third, in order to validate these results and assess the threshold of TIC characteristics in the future, a comprehensive prospective diagnostic investigation is required.

Conclusions

To sum up, our study’s findings showed that time-dependent measurements, such as arrival time, time to peak, arrival time change between the nodule and surrounding parenchyma, and time to peak change between the nodule and surrounding parenchyma, are crucial for differentiating between PTC that has a BRAF mutation and PTC that does not. Our research offers more proof that TIC-based quantitative metrics may help identify PTC patients’ BRAF V600E mutant status.

Supplemental Information

Supplemental Information 1 Raw data.

Supplemental Information 2 STROBE checklist.

Additional Information and Declarations

Competing Interests

The authors declare that they have no competing interests.

Author Contributions

Zhipeng Hu performed the experiments, prepared figures and/or tables, and approved the final draft.

Rong Xue performed the experiments, prepared figures and/or tables, and approved the final draft.

Zhixi Liu analyzed the data, authored or reviewed drafts of the article, and approved the final draft.

Liang Liu performed the experiments, analyzed the data, authored or reviewed drafts of the article, and approved the final draft.

Zheli Gong conceived and designed the experiments, authored or reviewed drafts of the article, and approved the final draft.

Human Ethics

The following information was supplied relating to ethical approvals (i.e., approving body and any reference numbers):

Medical Ethics Committee of Hunan Provincial People’s Hospital (The first Affiliated Hospital of Hunan Normal University) (IRB Approval No: [2024]-261)

Data Availability

The following information was supplied regarding data availability:

The raw measurements are available in the Supplemental File.

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
