# Peer review of "Role of contrast-enhanced ultrasound with time-intensity curve analysis about thyroid nodule and parenchyma for differentiating BRAF V600E mutation status"

_PeerJ, doi:10.7717/peerj.19006_

## Round 0.1 · original submission · Minor Revisions

· Academic Editor

Minor Revisions

Dear Dr. Gong,

The reviewers have commented on your manuscript. You can find the attached reports. Based on the comments and suggestions of the expert reviewers, a minor revision is needed for your article.

I request you check and correct the manuscript based on the reports.

Sincerely

Reviewer 1 ·

Basic reporting

Overall, professional English used throughout. There are a few areas where the grammar could be checked:
- there are discrepancies between the abstract given to reviewers and the one in the paper e.g. 'The BRAF V600E mutation was proved associate with Papillary' - I think this should read 'was proven associated with' or something similar? This is correctly written in the main text so may be a transcription error.
- Introduction, line 54 - there may be an omission (kinase mutation on exon...); can you specify the exon?
- Intro, line 59 - PTMCs is not an abbreviation that has been defined.
- Intro, line 67 'most studies were focus' - should this be focussed?
- Intro, line 64 - this needs a reference and also further clarification. Do you mean in the context of thyroid nodules or more generally? If thyroid nodules, you might consider this paper and discussing the test characteristics of CEUS (Trimboli, P., Castellana, M., Virili, C. et al. Performance of contrast-enhanced ultrasound (CEUS) in assessing thyroid nodules: a systematic review and meta-analysis using histological standard of reference. Radiol med 125, 406–415 (2020). https://doi.org/10.1007/s11547-019-01129-2)
- Image analysis, lines 101-102 - 'In addition, we measured more parameters which could get based on the above direct quantitative parameters' this needs rephrasing for clarity.

Experimental design

This is, overall, a thoughtfully designed study, given the limitations of retrospective research. The inclusion and exclusion criteria were clearly stated. The review process is a little unclear - were the reviewing radiologists the same as those performing the scans. The methodology could be improved by a blinded imaging review but it is not clear whether this was the case. This should be specified.

Validity of the findings

The data appear sound and robust. The conclusions drawn seem appropriate.

Additional comments

This was a concise study, within the limitation of a retrospective approach, with interesting findings.

Reviewer 2 ·

Basic reporting

1. the language is clear and unambiguous.
2. sufficient literature and background are provided.
3. Professional article structure, figures, tables. Raw data shared.
4. The paper is self-contained.

Experimental design

no comment

Validity of the findings

no comment

Additional comments

The current study analyzes the relationship between contrast-enhanced ultrasound (CEUS) with time-intensity curve analysis and the BRAF V600E mutation in thyroid tissue. The paper is concise and well-written. Below are my comments:

Major Points:
1. Sample Size: Did the authors include all available patients from October 2021 to December 2023? Please explain how the sample size was determined.
2. Background and Discussion: The authors elaborated on the relationship between BRAF status and the aggressive behavior of papillary thyroid carcinoma (PTC) in the background and discussion sections. While this emphasizes the clinical significance of the study, it also causes some confusion. In the discussion section, consider shortening the first paragraph and expanding the discussion in paragraph three. Given the numerous articles on the application of CEUS in thyroid cancer, a more comprehensive discussion of this literature would be beneficial.
3. Why do these parameters differ between the two groups? The authors could explore potential mechanisms underlying these differences in the discussion section.

Minor Points:
1. “One of the genetic mutations linked to the protein kinase pathway is the B-type Raf (BRAFV600E) kinase mutation on exon.” should be revised to: “One of the genetic mutations linked to the protein kinase pathway is the B-type Raf (BRAF) V600E kinase mutation on exon 15.”
2. Study Objective: Revise the sentence “In this study, we intended to explore not only the nodular parameters but also the quantitative parameters of the nodule and surrounding parenchyma, to provide more robust evidence for differentiating aggressive PTC and assisting clinical personalized treatment.” to: “In this study, we aimed to compare both the nodular parameters and the quantitative parameters of the thyroid nodule and surrounding parenchyma in relation to BRAF status. This study is expected to provide additional evidence for differentiating aggressive PTC and to assist in personalized clinical treatment.”
3. In Tables 2 and 3, mark p-values < 0.05 with asterisks.
4. In the discussion section, replace “retroactive” with “retrospective”.

---

## Round 0.2 · accepted · Accept

· Academic Editor

Accept

Dear Dr. Gong,

I thank you for making the corrections and changes requested by the reviewers. I read and checked your valuable article carefully and am happy to inform you that the article has been accepted for publication in PeerJ.
Sincerely yours

Reviewer 2 ·

Basic reporting

no comment

Experimental design

no comment

Validity of the findings

no comment

Additional comments

No further comments